# A Goal-Directed Program for Wheelchair Use for Children and Young People with Cerebral Palsy in Uganda: An Explorative Intervention Study

**DOI:** 10.3390/jcm12062325

**Published:** 2023-03-16

**Authors:** Carin Andrews, Angelina Kakooza-Mwesige, Sauba Kamusiime, Hans Forssberg, Ann-Christin Eliasson

**Affiliations:** 1Department of Women’s and Children’s Health, Karolinska Institutet, 171 76 Stockholm, Sweden; 2CURIE Study Consortium, Iganga-Mayuge Health and Demographic Surveillance System, Iganga P.O. Box 111, Uganda; 3Department of Pediatrics and Child Health, Makerere University College of Health Sciences, Kampala P.O. Box 7072, Uganda

**Keywords:** cerebral palsy, wheelchair, goals, intervention, low- and middle-income countries, participation

## Abstract

In this exploratory study, we investigate whether goal-directed intervention for wheelchairs can increase the activities of daily living for children and young people with cerebral palsy (CP) when implemented in rural Uganda. Thirty-two children and young people with CP (aged 3–18 years) participated in a home-visit intervention program, which included donating wheelchairs and setting individual goals. Goal achievement, frequency of wheelchair use, condition of wheelchairs, and caregivers’ perspectives were collected by interviews at 6–10 month after the start of intervention and the after three years. Our result show that most wheelchairs were in good condition and frequently used after 6–10 month with 83% goal achievement (132/158 goals; mean 4.3 (range 0–7). The caregivers reported several advantages (e.g., the child being happier) and few disadvantages (e.g., poor design and durability). At the three-year follow-up, only eleven wheelchairs were still used by 23 available participants (seven deceased and two moved). The children achieved 60% of their goals (32/53 goals mean 2.9; range 1–5). This demonstrates that the goal-directed intervention program for wheelchairs can be successfully implemented in a low-income setting with a high rate of goal achievement and frequent wheelchair use, facilitating participation. However, maintenance services are crucial to obtain sustainable results.

## 1. Introduction

Prescription of wheelchairs to individuals with mobility limitations has the potential to improve mobility, seating, and daily activities, and is an “essential component for inclusive sustainable development” [1,2,3]. In 2008, the World Health Organization (WHO) published guidelines for provision of wheelchairs in low- and middle-income countries (LMICs), emphasizing the need for a system addressing design, production, supply, and service delivery processes [4]. Application of comprehensive services has also shown promising results in some LMICs, resulting in higher use rates, higher user satisfaction, improved health, and improved performance in activities of daily living [5,6,7,8]. However, in most LMICs, provision of wheelchairs is still focused on the delivery of wheelchairs and not on the service delivery process, education, and training [2,9], resulting in dissatisfaction and poor usage [10,11].

Most studies on wheelchair usage in LMICs have covered a wide age range, while there are only few reports on children with mobility limitations. Because the needs of children differ in many aspects from adults, this provides a knowledge gap, and processes and guidelines developed for adults may not be optimal for children. A major problem is the limited access to wheelchairs in low-resource settings. Recent reports from Uganda, Bangladesh, and Vietnam show that only a small percentage of children with cerebral palsy (CP) requiring wheelchairs had received one [12,13,14]. This low access to wheelchairs in many LMICs is due to limited resources and because wheelchair provision is not included in national legislation, policies, or strategies [15]. The main mode of wheelchair provision in LMICs is through charitable donations, in which large numbers of wheelchairs are purchased and delivered to a target region [16]. Wheelchair donations are rarely accompanied by comprehensive education, training, and wheelchair service. Typically, a wheelchair is delivered at one point in time without formal assessment, fitting, training, or follow-up maintenance with support services [11]. The effects of such donations on wheelchair users are debatable, with some studies indicating low usage rates [10] and high rates of dissatisfaction [11], while other studies indicate that donated wheelchairs can lead to improved quality of life, health, and function [16,17].

We collaborated with a charity donating wheelchairs and utilized a previously identified population-based cohort of children and young people with CP [14,18]. We wanted to optimize wheelchair usage and increase the participation of the children in daily activities of the family and neighboring community, but could not find any appropriate program for the low-resource environment families were living in. Therefore, we developed a 6-month goal-directed intervention program based on family-centered care [19,20] in which we assumed that families needed support on how to incorporate wheelchairs into their daily activities because they have typically not seen any children use a wheelchair. In this exploratory study, we wanted to investigate the impact of goal-directed intervention on participation by studying goal achievement and how much the wheelchairs were used in daily activities, as well as family satisfaction.

## 2. Materials and Methods

### 2.1. Study Design

This was an explorative study with one group and a pre-post design, evaluating pre-defined goals and wheelchair use after a 6–10-month intervention period and at a three-year follow-up. The intervention comprised the donation of manual wheelchairs and family coaching and support, including home visits (Figure 1).

### 2.2. Study Setting

This study was performed at the Iganga Mayuge Health and Demographic Surveillance Site (IM-HDSS), which covers a population of around 90,000 people in 65 villages in a mainly rural area in Eastern Uganda [21]. Most of the population are engaged in subsistence farming. The population is served by two hospitals, sixteen community-based health centers, and a network of village health team workers providing community services, such as health promotion and empowerment in accessing and utilizing health services [21]. At the start of the study, there were no specialized services available for children and young people with disabilities and there were no systems in place for wheelchair provision, except for occasional charitable donations [14]. The typical home environment in the area comprised a cluster of homes around a compound, which is usually a mainly flat area of hard compacted soil where a lot of household activities, such as cleaning clothes and dishes and food preparation, take place (Figure 2).

### 2.3. Study Participants

Thirty-two (32) participants were recruited in 2016 from a population-based cohort of children and young people with CP identified through a three-stage screening process in the IM-HDSS [14,19]. The participants were 3–18 years old when the wheelchairs were distributed; all were non-walkers (GMFCS III-V); none had a wheelchair in working condition; and five (16%) previously had a wheelchair. Many participants needed assistance with manual activities (63%; MACS IV-V) and could not consistently communicate with familiar partners (72%; CFCS IV-V). Most families lived in rural areas, and most primary caregivers were mothers (81%). The caregivers had a primary school education or lower (81%) and worked as subsistence farmers (72%). Further characteristics are presented in Table 1.

### 2.4. Intervention Program

The intervention was intended to help families and children optimally use the donated wheelchair, through a goal-setting and coaching procedure upon delivery of the wheelchair and at two additional home visits one and two months later (Figure 1). The coaching approach aimed to increase the caregivers’ sense of self-efficacy and confidence in taking an active part in the implementation of goals [20,22].

### 2.5. Resources Needed for Implementation of the Intervention Program

The human resources needed for the implementation of the intervention program was a team of two therapists and one community mobilizer. The community mobilizer was responsible for communicating with the families and making sure that they were available to receive the team in their home at the agreed time. The therapists spent about one hour per child/per visit, which equals five hours per child for the implementation and evaluation. During each home-visit day, the team could meet 3–4 children, considering that travel time was about 2–3 h per day. One of the therapists (CA) was also responsible for coordinating the implementation program, which required additional work hours. The wheelchairs were donated, so they did not incur a cost in this program, but there was a need to buy some additional tools and equipment for adjustments, such as spanners, duct tape, and foam mattresses. We used a pick-up truck for wheelchair delivery at the first visit, and motorbikes at the subsequent visits to reduce costs.

### 2.6. The Donated Wheelchairs

The wheelchairs were donated by the charity Walkabout Foundation (https://www.walkaboutfoundation.org, accessed on 13 March 2023) through collaboration with the non-governmental organization Soft Power Health in Jinja, Uganda (https://www.softpowerhealth.org, accessed on 13 March 2023)). The families paid a symbolic sum of UGX 10,000 (about USD 3) for the wheelchair. The wheelchairs were constructed by the Association for the Physically Disabled of Kenya (https://www.apdk.org, accessed on 13 March 2023) and built for durability using a steel frame, solid rubber tires, and removable cushions made of foam, and were intended to be repairable and maintainable using local resources. The wheelchairs came in four sizes (i.e., 12, 14, 16, and 18 inches) and two models as follows: (I) a regular model and (II) an extra support model with a headrest, five-point harness, and extra supportive footrests. The wheelchairs often needed some individual fitting, which was conducted when delivered (Figure 3).

### 2.7. Goal-Setting Procedure

The goals were set in collaboration with caregivers and a therapist team (SK, local therapist; CA, Swedish therapist) at the first home visit when the wheelchairs were delivered (Figure 1). A family-centered and client-centered approach for goal setting was used; the key elements were listening, communicating, partnership, choice, and hope [19,20,22,23]. The therapists (SK is fluent in the local language of Lusoga) interviewed caregivers about the current situation and the daily activities of their child. By asking “What does the child or young people do in the morning, afternoon, and evening?”, followed by “Are there any other activities that your child does more rarely, etc. weekly, or monthly?”, this interview was used for goal setting, and the caregivers, therapists, and the child or young person (if they were able to) agreed on goals for wheelchair use. After the goals were decided, they discussed how these goals should be achieved and implemented during everyday situations. A weekly pictorial schedule of the targeted activity was jointly created and given to the caregivers to keep as a reminder.

At the second and third home visits, the therapist team used the same questions to follow up and to see if the goals were achieved and how often the wheelchair was been used. The caregivers and children were encouraged to show how they used the wheelchairs and discuss any difficulties encountered. The therapeutic coaching approach aimed to increase the caregivers’ sense of self-efficacy and confidence in taking an active part in the implementation of goals [20]. The weekly pictorial schedule was discussed and adjusted with new goals where appropriate. When needed, minor repairs and/or adjustments to the wheelchair were conducted using available resources, such as tools, foam, wooden blocks, and duct tape. Each visit lasted about one hour.

### 2.8. Assessments and Procedure

Data collection took place in the participants’ home environment by the same therapists who administered the intervention at 6–10 month after the start of the intervention and after three years (Figure 1).

A pre-structured questionnaire was developed, asking about the following: (A) wheelchair maintenance; (B) wheelchair use; and (C) wheelchair satisfaction (Appendix A). The questionnaire was developed by the research team together with local therapists working with children with CP in Uganda. It was developed to cover activities and topics highly relevant to this specific cultural setting. After development, the questions were tested on five caregivers of children with disabilities living in a similar area in Uganda. A Ugandan occupational therapist (SK) with knowledge of the local language Lusoga performed the interview.

For wheelchair maintenance (part A), the condition of the wheelchairs was inspected for need of repairs and some questions about repairs were asked.

The wheelchair use (part B) section contained 16 questions evaluating which activities were carried out and how often the wheelchair was used.

The wheelchair satisfaction (part C) part contained two open-ended questions where the caregivers were asked “What have been the advantages of getting the wheelchair?” and “What have been the difficulties encountered when using the wheelchair?” The therapist took notes during the interview and documented the answers immediately afterwards. This interview was only conducted at the first 6–10-month evaluation.

Achievement of goals: The goals set after the donation of the wheelchairs and additional goals set at subsequent two home visits were evaluated at the end of the intervention and at the three-year follow up. This was carried out by a therapist asking the caregivers whether the goal had been achieved or not with a “yes” or “no” answer.

### 2.9. Data Analysis

The data were double entered into Excel; any differences were corrected using the original data files. All goals defined at the start (136 goals) and the subsequent home visits (22 goals) were categorized in various topics and reported for daily, weekly, or monthly usage. The number of goal achievements was counted, and percentages were calculated by dividing the goal achievements by the number of set goals. Information on the use of the wheelchairs in various activities in the home, compound, and community was collected by semi-structured questions. The answers from the two open-ended questions on satisfaction were grouped into twelve categories for advantages and seven categories for disadvantages.

### 2.10. Missing Data

At the first evaluation, one participant missed data for all assessments (but was included in the 3-year follow up). At the 3-year follow-up, twenty-three participants were available (seven were deceased and two moved out of the area). Data on wheelchair use and goal for wheelchair were missing for two participants and data on goal achievement were missing for four participants.

### 2.11. Ethical Considerations

The study was conducted in accordance with the Declaration of Helsinki and approved by the Higher Degrees Research and Ethics Committee of the School of Public Health, College of Health Sciences, Makerere University and the Uganda National Council for Science and Technology (protocols HS 2608 and 1787, respectively). All caregivers provided written informed consent and assent was obtained from the children and young people where possible.

## 3. Results

### 3.1. Conditions before Intervention

At the start of the intervention, none of the children and young people had a functional wheelchair and all children at GMFCS level IV-V were dependent on being carried by caregivers. Only four of them had means of supported sitting, such as plastic chairs or locally made CP chairs, while the others lay down on the ground. Children at GMFCS level III could independently crawl shorter distances. Activity levels were low. Thirteen children and young people had no daily activities except eating/drinking and daily hygiene. Sixteen participants played daily with family and friends, eleven participants went to the mosque or church weekly or monthly, and ten were visited relatives or neighbors. No child or young people of school age went to school.

About half of the caregivers (*n* = 17) thought the symbolical sum of UGX 10,000 (USD 3) was affordable to pay for the wheelchair. They recognized that the fee was much smaller than the actual cost, and that the wheelchair was important for their child. The others did not think it was affordable but all except two managed the cost (these two families received a private donation to cover the cost). Due to low income (subsistence farmers), they had to sell livestock or agricultural goods to be able to pay the fee.

### 3.2. Evaluation at 6–10 Months

#### 3.2.1. Achievement of Daily, Weekly, and Monthly Goals

The daily, weekly, and monthly goals are presented in Table 2, together with goal achievement rate. Of the total 158 goals, 132 (83%) were achieved. Each child achieved, on average, 4.3 goals (range 0–7). Daily activity goals were most common and 87% of these were achieved. Playing with other children in the compound was the most common goal achieved, followed by eating or being fed while sitting in the wheelchair, being pushed around by family members on walks, and being put in a sitting position. The weekly goals were achieved by 84% of the participants, and the wheelchairs were used for participation in religious services, being on the fields with family members, and visiting relatives and neighbors. Of the monthly goals, 63% were achieved. The most common goal was related to transport to healthcare when sick and going to parties. Thirteen children who did not have any other daily activity apart from eating and daily hygiene before they obtained the wheelchair achieved 3.8 goals (range 0–6), resulting in a 79% goal achievement rate.

#### 3.2.2. Interview of Wheelchair Satisfaction

The responses to the interview on the advantages of using the wheelchair were sorted into twelve categories (Table 3). They were often related to the child’s mood, i.e., the child was happier, had increased interaction, and was able to experience new environment. One caregiver said the following: “it has changed her life totally; she has made new friends and she is always happy” (aunt of an 8-year-old girl, GMFCS III). The wheelchairs increased participation in the home and in community activities, such as being able to play with other children and young people and to join family members on trips to different places. Social interactions increased by allowing them to make new friends and experience new environments at greater distances. Only two children could manually operate the wheelchairs on their own, but other children would often push the wheelchair. The wheelchairs made a dramatic difference for the children who were previously only able to stay indoors lying on the ground all day; now they had the possibility to sit up and be moved around the compound. Caregivers also noted that the children’s motor function improved, for example, stronger trunk and neck muscles, improved hand function when sitting, and being able to walk using the wheelchair for support. The most important advantage from caregivers’ perspective was not having to carry the child from place to place, which eased the physical burden. The wheelchair allowed other people, such as siblings, to help move the child around, and made it easier for caregivers to bring the child with them to different places, such as when working in the courtyard.

Six categories regarding difficulties when using the wheelchair were reported (Table 3). More than half of the caregivers (19/31) reported that they had not encountered any difficulties. The most common difficulties were related to wheelchair design and durability, e.g., insufficient head support or parts of the wheelchair being worn out.

#### 3.2.3. Frequency of Wheelchair Use

A majority (81% *N* = 26/32) of the children and young people used their wheelchairs daily and three children used them only weekly, while two children did not use them at all due to sickness. In terms of daily use, the wheelchair was mainly used at home, in the compound (close area around the houses), and in the neighborhood for sitting, moving around, and quiet leisurely activities (Table 4). In terms of weekly use, the wheelchair was commonly used in the neighborhood and the community to move around, shopping, errands, and religious activities. In terms of monthly use, the wheelchair was used for transport to health centers, following caregivers to workplaces, and social activities (Table 4).

#### 3.2.4. Wheelchair Maintenance

Twenty-nine of the thirty-one donated wheelchairs were in use during the first 6–10 months, while two had not been used due to the poor health condition of the children. Sixteen wheelchairs were well functioning and five had been repaired, while ten needed some repairs, although they were still in use. The parts in need of repairs were the wheels (5), brakes (1), table (4), footrest (3), and harness (1). The reasons for not repairing the wheelchair were that the family had not been motivated to conduct the repairs, they did not have money to do, or they were not sure of how to fix or to find someone that could repair it.

### 3.3. Evaluation after Three Years

It was only possible to find 23 of the 32 participants after three years. Seven children were deceased [24] and two participants had migrated from the area and could not be found. Goal achievement data were missing for two children, and one child was excluded because she had learnt to walk and did not use the wheelchair anymore (*N* = 20).

#### 3.3.1. Goal Achievement of Wheelchair Use

The 20 participants with goal achievement data after three years had set 106 goals during the intervention (see Table 2). Thirty-two goals (30%) were achieved (average 1.6; range 0–5 goals/person). Goal achievement for daily use was 30%; this figure was 28% for weekly use and 35% for monthly use. A much better goal achievement rate was reached among the eleven children still using their wheelchairs. They achieved 32 (60%) of their 53 goals (average 2.9; range 1–5). Goal achievement for daily use was 61%, 50% for weekly use and 75% for monthly use.

#### 3.3.2. Frequency of Wheelchair Use

A total of 13 of the 23 participants still used their wheelchairs. Of those, 12 used it every day, and 1 used it only monthly. The wheelchairs were mainly used for sitting, moving around, and leisure activities in the home, compound, and neighborhood (Table 4), Ten participants did not use the wheelchair because it was broken (4), too small (2), or had been left at their grandparents’ home (1). Three children had usable wheelchairs but did not use them because they were unhappy in the wheelchair (2) or had started walking (1).

#### 3.3.3. Wheelchair Maintenance

Twenty-one wheelchairs were inspected because one had been stolen and one had been left at another family member’s home. Eight wheelchairs were in good condition and one of them had been repaired. The remaining 13 wheelchairs needed repair, e.g., wheels, harness, brakes, backrest, table, headrest, cushions, seat, and footrest. The reasons for not repairing them was a lack of money, not finding someone that could repair it, or not finding spare parts or tools.

## 4. Discussion

The goal-directed program promoting daily wheelchair usage was easy to perform across families and led to a high rate of goal achievement. The achievements of goals led to frequent use of the wheelchair and increased participation in daily activities. Caregivers were overall very satisfied and mentioned many advantages of the wheelchairs. In general, the follow-up three years after the wheelchair donation exposed the weaknesses of not providing long-term services for wheelchairs because only eight were in good condition, and only thirteen of the remaining twenty-three children were still using the wheelchairs. However, these children still had a high rate of goal achievements, considering the long time-frame.

### 4.1. The Wheelchair-Program—Easy to Implement and Affordable

This wheelchair program focused on one step (education and training) in the WHO eight-step guidelines for the prescription of wheelchairs in low-resource settings [4]. We had previous information regarding children and young people with CP [14] complying to the three first steps (referral, assessment, and prescription). Donated wheelchairs were available for the participants (step 4), but there was limited opportunity to select and adjust the wheelchairs (steps 5 and 6). This study focused on step seven, developing and evaluating a training program with a coaching approach. These families, living in rural areas, had typically not seen other children use wheelchairs. As such, our assumption that the families needed guidance and support on how and when to use the wheelchair for their specific needs and their environment seemed right, and the family and client-centered practice approach was helpful [19,20] in the goalsetting procedure. In addition, previous experience of contact with the families was important as it helped us to make the goal-setting process as concrete as possible by asking simple questions, such as “what does your child do in the morning, afternoon and evening?”. We also found subsequent home visits motivated the families to continue to work towards increased use of the wheelchair and to possibly set up new goals. We have not previously seen goal-setting approaches applied in programs promoting wheelchair usage, and there are also limited studies using goal-directed interventions in LMICs [25]. Our study suggests that this goal-directed program is effective, feasible, and easy to implement in a low-resource setting with minimal resources, with just a few home visits from therapists.

### 4.2. High Rate of Goal Achievement Resulted in Frequent Wheelchair Use and Increased Participation

Overall, there is little information in the research literature on training programs for optimal wheelchair usage in children with mobility limitations, particularly for children living in low-resource settings [7,26]. The only broadly spread wheelchair program we found is a skill-training program for adults (https://wheelchairskillsprogram.ca, accessed on 13 March 2023). Because the aims for the prescription of the wheelchairs often differ between adults (after injuries) and children with developmental disabilities, such skill-training programs are not appropriate for children with CP who, due to their often severe functional limitations (and, in this case, the surrounding terrain), are prevented from driving wheelchairs themselves. For many of these children, a main gain is the opportunity to sit up and take part in daily activities and to be moved around in order to participate in various activities in the family and community. The results from this study showed a very high rate of goal achievement of daily, weekly, and monthly goals, leading to frequent use in the home and community. Before the intervention, about half of the children and young people had no other daily activities other than eating and maintaining hygiene. In fact, they would often spend all their time in one place, often lying down on a mattress or mat inside the house, unable to change position or environment, which led to exclusion and isolation. We did not specifically measure participation (for example, see [27]), but almost all goals implicated participation in various activities. The caregivers also identified several advantages of the wheelchair and few disadvantages. By using the wheelchair, the participants’ life situations were dramatically changed. Caregivers described that the children and youth were happier, and the wheelchairs made it possible to bring them to different places and experience new environments. From a caregiver perspective, a wheelchair provided great relief. Their workload decreased because they previously had to carry the child for all kinds of transfers. The burden of carrying a child with disabilities can lead to musculoskeletal pain and exhaustion for caregivers in low-income settings [28,29].

### 4.3. Maintenance Issues a Challenge

After three years, only 13 children used their wheelchairs out of the 23 children available at the follow-up because the wheelchairs were broken or did not fit the child anymore. This is not unexpected as studies from North America found that more than half of wheelchairs require repairs within 6 months [30] and that brakes, seats, footplates, and casters deteriorate before the expected three-year lifespan of a wheelchair [31]. Considering the additional influence of a tropical environment and rough terrain wearing down the wheelchairs, it is rather surprising that so many wheelchairs were still in use. A big problem, however, was that the wheelchairs were donated without any provision of maintenance services and that most families had no or limited capacity to repair the wheelchairs themselves. This is a well-known problem in low-resource settings. In a study from India, only 18% of donated wheelchairs were used, while the rest were left without use or were sold [10]. Likewise, a study from Zimbabwe including children with CP showed high levels of dissatisfaction with wheelchair features and services, requesting a minimum standard for services when donating wheelchairs [11].

Our findings corroborate the importance of continued support for donated wheelchairs and that maintenance of the wheelchairs and the replacement of parts or the whole wheelchair is needed within regular intervals, in accordance with the WHO wheelchair step eight on maintenance and support. Consequently, our results showed that wheelchair use was low after three years. Yet, when only studying the children still using their wheelchairs, goal achievement was high, underscoring the importance of a comprehensive wheelchair program, including education, training, and maintenance.

### 4.4. Limitations

A strength of this report is that it is one of the first to address the problem of donating wheelchairs to children in low-resource settings where people lack experience on how to use them. This is an exploratory study on the feasibility of a newly developed goal-directed intervention program promoting daily use. A limitation of the present study is the limited number of participants, as well some missing data. Additionally, the study design without a control group made it possible to only study use after combined intervention and wheelchair donation. However, compared to the limited information in the literature [10,11], the usage in our cohort was outstanding in the short term and was still very high after three years among those who still had a usable wheelchair.

We chose to use parents’ reports instead of observation and this may be viewed as a limitation. This might have led to a social desirability bias with the caregivers wanting to answer what they thought we wanted to hear. We tried to mitigate this by asking the caregivers to explain how and when the wheelchair was used and always asked to see the child in the wheelchair to see the condition of the wheelchair. On the other hand, caregivers’ perspectives are crucial in a client-centered approach to develop recommendations to improve health [32]. It would also have been nice to include a quality-of-life questionnaire and a stress index for caregivers because we could only indirectly measure these dimensions in the present study. We did not conduct a cost-effectiveness analysis, but we consider additional home visits from the therapist team as affordable in the long run.

## 5. Conclusions

This study shows that donation of wheelchairs in combination with an intervention program profoundly increased mobility and participation in daily activities for children and young people with cerebral palsy in a low-resource setting. The program was based on goal-setting principles in which caregivers, in collaboration with therapists, identified daily, weekly, and monthly activities in which the child should use the wheelchair. A high degree of goal achievement led to frequent use of the wheelchairs and increased participation both in the family and community.

## Figures and Tables

**Figure 1 jcm-12-02325-f001:**
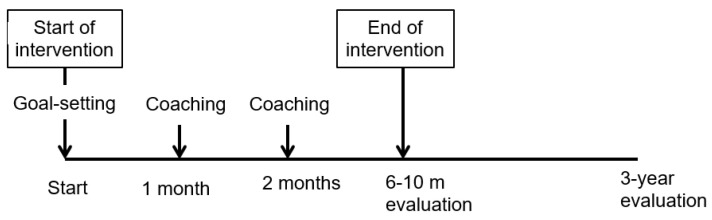
Timeline of the intervention and evaluation in a total of five home visits. Data collection occurred at three points (start, after 6–10 months, and 3-year follow-up).

**Figure 2 jcm-12-02325-f002:**
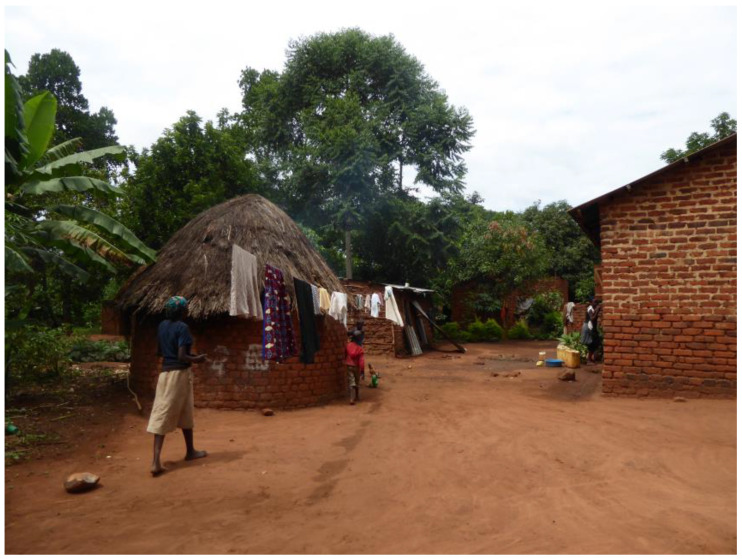
A typical compound in this rural area of Uganda.

**Figure 3 jcm-12-02325-f003:**
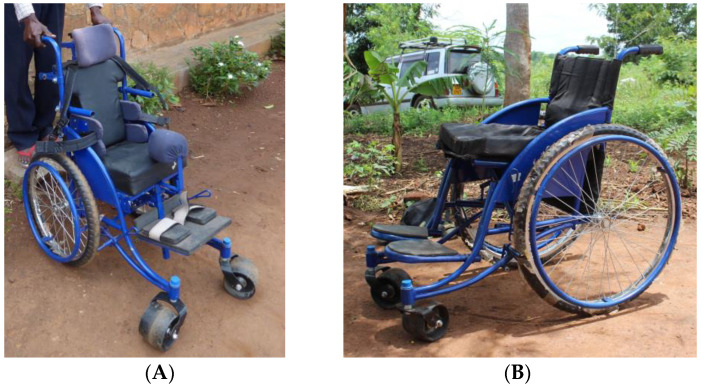
The two different types of wheelchairs used in the program: (**A**) Extra support model with a headrest, five-point harness, and extra supportive footrests and (**B**) regular model.

**Table 1 jcm-12-02325-t001:** Background information for the participants and their main caregivers.

Background Information Children/Young People		*N* = 32
Age at wheelchair distribution	3–5 years	12
	6–11 years	15
	12–18 years	5
Gender	Female	15
	Male	17
Residence area	Rural	23
	Semi-urban	9
GMFCS level	III	17
	IV	4
	V	11
MACS level	I/II	1/3
	III	8
	IV/V	7/13
CFCS level	I/II	1/1
	III	7
	IV/V	4/19
**Background Information of Main Caregivers**		***N* = 32**
Relationship to child	Mother/Father	26/2
	Grandmother/Sister	3/1
Education	None/Missing	4/1
	Primary	22
	Senior/Tertiary	4/1
Occupation	Subsistence farmer	23
	Petty trading	3
	Other/Missing	5/1

**Table 2 jcm-12-02325-t002:** Goals set and achieved for different areas of wheelchair use, presented for daily, weekly, monthly or few times/year at 6–10-month and 3-year follow-ups. The 3-year follow up is reported for all children possible to follow up (*N* = 20) and those that used the wheelchair (*N* = 11).

Daily Activity Goal	6–10 Month		3-Year Follow-Up			
	Goal Set/Achieved 31 Children		Goal Set/Achieved 20 Children		Goal Set/Achieved Wheelchair Users *N* = 11	
	*N*	%	*N*	%	*N*	%
Playing with other children in compound	27/26	96	16/8	50	9/8	89
Being able to eat/feed in the chair	15/12	80	10/1	10	3/1	33
Being pushed around by family members for walks	14/13	93	11/5	45	9/5	56
Be in a sitting position	14/13	93	10/3	30	4/3	75
Practicing self-driving and transfers	10/8	80	6/0	0	4/0	0
Going to the fields with family members	5/4	80	3/1	33	1/1	100
Visiting relatives and neighbors	4/3	75	3/1	33	1/1	100
Walking practice using wheelchair as a walker	3/1	33	1/0	0		
Self-driving around compound	2/2	100	1/0	0		
Going to shops	2/2	100	2/0	0		
Being outside and sunbathing	2/2	100	1/0	0		
For transport to mothers’ workplace	2/2	100	0	.		
Going to school	1/0	0	0	.		
**Total daily activity goals**	**101/88**	**87**	**64/19**	**30**	**31/19**	**61**
**Weekly activity goals**	**Goals Set/Achieved** **31 Children**		**Goals Set/Achieved** **20 Children**		**Goal Set/Achieved** **Wheelchair Users *N* = 11**	
	** *N* **	**%**	** *N* **	**%**	** *N* **	**%**
Visiting the church or mosque	22/20	91	14/5	36	8/5	63
Going to the fields with family members	6/4	67	6/1	17	2/1	50
Visiting relatives and neighbors	7/5	71	5/1	20	4/1	25
Going to shops	1/1	100	0	.		
Being pushed around by family members for walks	1/1	100	0	.		
Going to the swimming pool	1/1	100	0	.		
**Total weekly goals**	**38/32**	**84**	**25/7**	**28**	**14/7**	**50**
**Monthly goals or goals for few times/year**	**Goal Set/Achieved** **31 Children**		**Goal Set/Achieved** **20 Children**		**Goal Set/Achieved** **Wheelchair Users *N* = 11**	
	** *N* **	**%**	** *N* **	**%**	** *N* **	**%**
Transport to healthcare when sick	11/6	55	9/4	44	5/4	
Going to parties	6/5	83	5/1	20	1/1	
Going to visit relatives and neighbors	0	0	1/0	0		
Going to visit the family village	1/0	0	1/0		1/0	
Going to barber	1/1	100	1/1	100	1/1	
**Total monthly or few times/year goals**	**19/12**	**63**	**17/6**	**35**	**8/6**	75
**Total, all goals**	**158/132**	**83**	**106/32**	**30**	**53/32**	**60**

**Table 3 jcm-12-02325-t003:** Categories of advantages and disadvantage of using the wheelchair as identified by caregivers at the 6–10-month evaluation; each caregiver (*N* = 31) could identify several advantages, resulting in more advantages than participants.

Advantages of Using the Wheelchair	*N*
Child is happier	17
Do not have to carry the child	15
Improved social and play interactions with other children	15
Child gets to see new places and experience different environments	14
Improved motor function	14
Child can change position	11
Easier to bring child to different places	8
Easier feeding	4
Improved cleanliness of the child	3
Easier for child to attend school	2
Faster and safer mobility	1
Child can fetch water	1
**Disadvantages of Using the Wheelchair**	
Wheelchair construction problems	6
Difficulties feeding the child in the wheelchair	3
Swollen feet	2
Too sick to be sitting	1
Mother too busy to help the child	1
Concerns about the safety of sitting in the wheelchair because of convulsions	1

**Table 4 jcm-12-02325-t004:** Number of children using the wheelchairs for different purpose after 6 months *N* = 31 and 3 years *N* = 23 at home, in the compound (the close area around the houses), and in the community.

	Daily		Weekly		Monthly		Not Used	
	6–10 Months	3 Years	6–10 Months	3 Years	6–10 Months	3 Years	6–10 Months	3 Years
Wheelchair use in the home and compound								
Sitting inside home	9	2	3	0	0	0	19	21
Moving inside the house	2	0	2	0	0	0	27	23
Sitting outside in the compound	16	8	5	1	0	1	10	13
Moving around the compound	24	8	5	1	0	0	2	14
Quiet leisure activities	23	5	5	4	0	0	3	14
Eating/feeding	12	3	3	1	0	0	16	19
For housework	0	2	0	1	0	1	31	19
Wheelchair use in the community								
Moving around the neighborhood	12	6	13	3	1	1	5	13
For agricultural activities	1	1	5	2	5	0	20	20
Shopping and errands	0	1	11	2	4	0	16	20
Transport to health center	0	0	0	0	19	7	12	16
Transport to caregivers working place	0	0	0	0	19	6	12	17
Transport to working place/school	3	0	4	0	1	0	23	23
At working place/school	3	0	1	0	0	0	27	23
Taking part in social activities	0	1	0	2	13	4	18	16
Taking part in religious activities	1	0	17	6	4	1	9	16

## Data Availability

Data sharing is not applicable due to privacy and ethical restrictions.

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
