# Peer review of "A Goal-Directed Program for Wheelchair Use for Children and Young People with Cerebral Palsy in Uganda: An Explorative Intervention Study"

_jcm, 2023, doi:10.3390/jcm12062325_

Round 1

Reviewer 1 Report

Very interesting topic and one that deserves attention and research. Please find attached document on suggestions. 

Line 57 sentence is awkward and not understanding the meaning

Line 58 states “studies” but only cites 1 study

Line 61 citation does not match the others

Line 66 – how did you arrive at your assumptions? Was there conversatios/interviews with local therapists/families/ etc?

Line 89 need [

Line 91 add word “in”

Line 92 consider changing people to young adults so it reads children and young adults?

Line 93 – citation deos not match

Line 95 consider removing first had

Line 98 – consider revising- it reads as though the children had a primary education and worked on the farms.

In the table you reported a subject being MACS level I is that accurate?

Would be nice to include a picture of the two different types of wheelchairs donated.

Were the therapists local to Uganda? Was translation used?

Line 140- each visits to be changed to each visit

Line 202 consider revising the word without to make more understandable

In Table 2 I do not understand why there are * in the table

Can you give some description of what is considered the compound? Because you discuss the community below

Line 241 fix advantages- should not be capitalized

Line 251 consider revising sentence for clarity

Line 275 citation does not match

Line 302 of discussion you state the program was easy to implement, however, you did not really describe that process or interview the therapists or families on the ease of implementation of the program

Author Response

Line 57 sentence is awkward and not understanding the meaning

Line 58 states “studies” but only cites 1 study

Line 61 citation does not match the others

Reply: Sorry for the mismatch between ref, all language correction is inserted and ref checked

Line 66 – how did you arrive at your assumptions? Was there conversatios/interviews with local therapists/families/ etc?

Reply: It is based on previous population based studies where we meet the families for other reason. We know that most families has never meet other children in wheelchair and we know the user rate is low. A clarification is inserted in the text.

Line 89 need [

Line 91 add word “in”

Reply: adjusted

Line 92 consider changing people to young adults so it reads children and young adults?

Reply: We have had several discussions on which term to use for this group, teenager, young adults, or youths and so on. This article is one of several studies from this cohort population-based cohort. After previous discussions with editors we have used the word young people in previous studies, therefore we prefer to use it also for this study.

Line 93 – citation deos not match

Line 95 consider removing first had

Reply: adjusted

Line 98 – consider revising- it reads as though the children had a primary education and worked on the farms.

Reply: The sentence is adjusted

In the table you reported a subject being MACS level I is that accurate?

Reply: Yes this is correct, one of the children at GMFCS level III had a MACS level I.

Would be nice to include a picture of the two different types of wheelchairs donated.

Reply: We have added two pictures

Were the therapists local to Uganda? Was translation used?

Reply: The therapists were one from Uganda (SK) and one from Sweden (CA), SK is fluent in the local language Lusoga, and CA were familiar to the local context since she lived and worked in the area for four years prior to the study. SK did all the interviews and goal-setting with the families, while CA were present and were part of the interviews and discussions. We have added some information on this under the heading 2.6 Goal setting procedure

Line 140- each visits to be changed to each visit

Reply: adjusted

Line 202 consider revising the word without to make more understandable

In Table 2 I do not understand why there are * in the table

Reply: It was aimed to clarify the number of children included in each group, however that was probably more confusing than clarifying, so it is now deleted.

Can you give some description of what is considered the compound? Because you discuss the community below

Reply: The typical home environment comprises of a cluster of homes around a compound. The compound is usually a mainly flat area of hard compacted soil, where a lot of the household activities take place, such as cleaning dishes and clothes, preparation of food etc. See attached photo of a typical compound. Although the compound is accessible for people from the wider community, it is considered part of the family’s home area. I have added some information on the compound under heading 2.2 Study setting.

Line 241 fix advantages- should not be capitalized

Line 251 consider revising sentence for clarity

Line 275 citation does not match

Reply: adjusted

Thank you very much for an clear and easy so understand review.

Line 302 of discussion you state the program was easy to implement, however, you did not really describe that process or interview the therapists or families on the ease of implementation of the program

Reply: You are right, we have not done a proper evaluation of whether it was easy to implement. We have changed the word implementation to perform.

Reviewer 2 Report

This is a prospective longitudinal report of 32 Ugandan families whose children with cerebral palsy received donated wheelchairs and were included in a goal-setting program. All families were followed for 6-10 months, and 2/3 of the families were captured again at a 3-year follow-up. Results were very positive.

There is little or no similar research, and so this exploratory study is ground-breaking. It points the way to service providers regarding the benefits and challenges of donating wheelchairs in LMID countries and it provides a foundation for more rigorous research in this area. The report is clear, succinct and well written.

I have divided my report into main points and minor points. However, there are no major revisions, and I am happy to leave the decisions about addressing these points to the researchers.

MAIN POINTS

In the introduction, there is very previous literature about outcomes of providing wheelchairs to adults or children in LMICs. Is that because the literature is lacking? Or is it possible to give more background here?

The sentence “all were non-independent walkers” makes it sound that all walked. But if their GMFCS ranged from III to V, then presumably some were completely non-ambulant. So would it be clearer to say something like “none could walk unassisted (GMFCS III-V)”? If you know how many were completely non-ambulant at the start of the study, then please indicate this.  

A query about Section 2.6. It says: “Through asking: “What does the child or young person do in the morning, afternoon, and evening?” followed by: “Are there any other activities that your child does more rarely, etc. weekly, or monthly?” *** Thereafter the caregivers, therapists, and the child or young person (if they were able to) agreed on where and when the wheelchair should be used. This was followed by discussions on how these goals should be achieved and implemented during the everyday situation.” There seems to be step missing where I’ve put three asterisks***. After asking what they did frequently and rarely, presumably you asked them what they wanted to be able to do or what were their goals? Only after setting goals could you talk to them about “how these goals should be achieved and implemented”.

Thank you for including the exact questionnaire in the appendix. If you hadn’t, I would have asked for it.

The results are very clearly presented. I make one tentative suggestion (but am happy to leave the final decision to the authors): I would change the order of the sections: (1) achievement of goals (because this was your primary aim), (2) satisfaction and difficulties (because that seems to follow naturally from goals), (3) frequency of use, (4) maintenance and repairs (I would put this last).

Can the authors provide any information on the time required or costs involved in delivering this program?

In the limitations section, the authors could suggest that future research in this area might investigate the effects on child’s quality of life. Their own results would suggest that this was improved, although a quality of life scale was not used in this study. They also noted anecdotally the effects on caregivers, and so a measure of caregiver stress or family functioning or family quality of life would be useful in any similar research.

MINOR POINTS:

In the abstract: Change “activities of daily living by implementing for children” to “activities of daily living by implementing for children”.  

Where you say, “low usage rates10 and high rates of dissatisfaction 11 while,” did you intend to add references where these numbers appear?

Change “participants were recruited 2016” to “participants were recruited in 2016”.

Please define this abbreviation the first time it appears: IMHDSS.

Change “participants home environment” to “participants’ home environment”.

Change “with local therapist” to “with a local therapist”.

Change “reparation” to “repairs”. (Reparation means compensation for injury.)

Change “all without two manage the cost” to “all except two manage the cost”.

In the title to Table 2, change “Numbers of children using the wheelchairs for different purpose after 6 months N=31* and 3 ** years N=23” to “Numbers of children using the wheelchairs for different purpose after 6 months (n=31) and 3 years (n=23) organized in two groups”. NB: I suggest omitting “organized in two group” because they are reported all as one group, only across 2 time points. And this is clear enough without adding it to the title of the table. The same applies to the title of Table 3.

In the title to Table 3, change “for i.e., daily, weekly” to “for daily, weekly”.

In Table 4, change “To sick to be sitting” to “Too sick to be sitting”. And left-justify the first column (which is currently centred).

Author Response

Thank you for a very clear and perceptive review.

MAIN POINTS

In the introduction, there is very previous literature about outcomes of providing wheelchairs to adults or children in LMICs. Is that because the literature is lacking? Or is it possible to give more background here?

Reply: There is very little available literature of wheelchair provision for children in LMICs. We think we have covered most of the literature available after searching on pubmed.

The sentence “all were non-independent walkers” makes it sound that all walked. But if their GMFCS ranged from III to V, then presumably some were completely non-ambulant. So would it be clearer to say something like “none could walk unassisted (GMFCS III-V)”? If you know how many were completely non-ambulant at the start of the study, then please indicate this. 

Reply: We are no using the word non-walker since no children had any walking ability but some were crawling.

A query about Section 2.6. It says: “Through asking: “What does the child or young person do in the morning, afternoon, and evening?” followed by: “Are there any other activities that your child does more rarely, etc. weekly, or monthly?” *** Thereafter the caregivers, therapists, and the child or young person (if they were able to) agreed on where and when the wheelchair should be used. This was followed by discussions on how these goals should be achieved and implemented during the everyday situation.” There seems to be step missing where I’ve put three asterisks***. After asking what they did frequently and rarely, presumably you asked them what they wanted to be able to do or what were their goals? Only after setting goals could you talk to them about “how these goals should be achieved and implemented”.

Thank you for including the exact questionnaire in the appendix. If you hadn’t, I would have asked for it.

Reply: Thank you for noticing this, we have added a sentence.

The results are very clearly presented. I make one tentative suggestion (but am happy to leave the final decision to the authors): I would change the order of the sections: (1) achievement of goals (because this was your primary aim), (2) satisfaction and difficulties (because that seems to follow naturally from goals), (3) frequency of use, (4) maintenance and repairs (I would put this last).

Can the authors provide any information on the time required or costs involved in delivering this program?

Reply: That is a good suggestion and the order is adjussted in the manuscript. It is also a good suggestion to include resources needed for the programme, since this is important information for future implementation, we have added a heading for this “2.5 Resources needed for implementation of the intervention programme”

In the limitations section, the authors could suggest that future research in this area might investigate the effects on child’s quality of life. Their own results would suggest that this was improved, although a quality of life scale was not used in this study. They also noted anecdotally the effects on caregivers, and so a measure of caregiver stress or family functioning or family quality of life would be useful in any similar research.

Reply: Thank you for this suggestion, now inserted in the manuscript.

MINOR POINTS:

In the abstract: Change “activities of daily living by implementing for children” to “activities of daily living by implementing for children”. 

Reply: adjusted

Where you say, “low usage rates10 and high rates of dissatisfaction 11 while,” did you intend to add references where these numbers appear?

Reply: Sorry, this is references but missing the brackets

Change “participants were recruited 2016” to “participants were recruited in 2016”.

Please define this abbreviation the first time it appears: IMHDSS.

Change “participants home environment” to “participants’ home environment”.

Change “with local therapist” to “with a local therapist”.

Change “reparation” to “repairs”. (Reparation means compensation for injury.)

Change “all without two manage the cost” to “all except two manage the cost”.

Reply: the changes are done accordingly

In the title to Table 2, change “Numbers of children using the wheelchairs for different purpose after 6 months N=31* and 3 ** years N=23” to “Numbers of children using the wheelchairs for different purpose after 6 months (n=31) and 3 years (n=23) organized in two groups”. NB: I suggest omitting “organized in two group” because they are reported all as one group, only across 2 time points. And this is clear enough without adding it to the title of the table. The same applies to the title of Table 3.

In the title to Table 3, change “for i.e., daily, weekly” to “for daily, weekly”.

In Table 4, change “To sick to be sitting” to “Too sick to be sitting”. And left-justify the first column (which is currently centred).

Reply: the changes are done accordingly